# The Potential of Wind Power-Supported Geothermal District Heating Systems—Model Results for a Location in Warsaw (Poland)

**Bartłomiej Ciapała** [1,*] **, Jakub Jurasz** [2,3] **and Alexander Kies** [4]

[1] Department of Fossil Fuels, Faculty of Geology, Geophysics and Environmental Protection, AGH University of Science and Technology, 30-059 Kraków, Poland

[2] Department of Engineering Management, Faculty of Management, AGH University of Science and Technology, 30-059 Kraków, Poland; jakub.jurasz@mdh.se

[3] School of Business, Society and Engineering, MDH University, 722 20 Västerås, Sweden

[4] Frankfurt Institute for Advanced Studies, Goethe University, 60438 Frankfurt, Germany; kies@fias.uni-frankfurt.de

\* Correspondence: bciapala@agh.edu.pl

**Abstract:** Geothermal heat is considered a sustainable energy source with significant global potential. Together with heat distribution networks, it can provide clean thermal energy to individual and commercial consumers. However, peaks in heat demand can require additional peaking sources at times. In this paper, we investigated how wind turbines can act as a peak energy source for a geothermal district heating system. We studied a model consisting of a geothermal heat source, a heat storage and wind power generator using historical weather data of Warsaw (Poland) and showed that wind power could increase the renewable share to supply a considerable heat demand compared to a geothermal heat source alone. The results indicate that wind power can be a suitable complement for a geothermal heat source to provide energy for heating. It is shown that a theoretical geo-wind-thermal storage based district heating network supplying 1000 m$^2$, which requires 100 W/m$^2$ at an outdoor temperature of $-20$ °C should have the following parameters: 4.8 MWh of thermal energy storage capacity, 45 kW of geothermal capacity and 5 kW of wind capacity. Such a system would ensure minimal wind curtailment, high utilization of geothermal source and high reliability of supply.

**Keywords:** district heating; geothermal; wind power; hybrid energy source; renewable balancing

## 1. Introduction

Geothermal district heating (GeoDH) systems using hot geothermal water usually require large investments and are constrained not only from investment and environmental conditions but also from the properties and availability of geothermal reservoirs. The last requirement is often the crucial limitation of geothermal heating plants based on a given number of wells. Limitation of heat capacity is especially important for low-temperature geothermal reservoirs, where only geothermal water extraction is possible (i.e., the vast majority of prospective and commonly accessible Mesozoic sedimentary basins extending within Central Europe). Facing such constraints, decision-makers have decided to establish district heating (DH) in order to not abandon this reliable and relatively cheap renewable source of energy. In such DH networks the baseload is provided by a geothermal plant, while comparatively rarely used peak power is supplied by another source of energy.

The sustainably established geothermal heat source is a steady and reliable energy source [1], with its power limited by local geological conditions. However, it is difficult to increase the capacity of

geothermal wells and drilling additional ones is not always a possible solution. Therefore, the need for a peak heat source [2] emerges. Wind power, on the contrary, fluctuates on short and medium timescales, which can lead to significant curtailment [3] due to supply/demand mismatch and limited storage potential. However, scaling wind farms is much easier and therefore wind sources are better suited for supplying the peak demand than geothermal wells. At the same time, both sources are characterized by relatively low cost per unit of energy [4].

Geothermal energy is known for its stability, although some output fluctuations are common, as well as a slight decrease of reservoir parameters (pressure, temperature) over a decades-long period [5]. Both factors do not significantly influence the short-term operation, but should be included in the peak power demand sizing during the investment stage.

Meanwhile, wind power does not only vary significantly within seconds (gusts) but often also on a seasonal and multi-year scale leading to some uncertainty [6,7], which should also be considered during the design phase.

Geothermal-powered district heating systems are usually backed with a non-geothermal peak-demand heat source. Despite their comparatively high rated power, they often produce a disproportionately small share of energy [8,9] and solely satisfy peak power demand. There are several typical heat sources used in geothermal district heating systems to provide peak power, the most popular ones being natural gas boilers [2] and natural gas-fueled cogeneration units. Another option includes entirely renewable-powered geothermal district heating systems, that use biomass as a fuel for peak-power boilers [10].

Simultaneously, hybridisation is a recognised method of ecological enhancement in the energy field [11], allowing for a reliable and multi-sourced district heating system operation [12].

Coupling wind power with the heating system is a known concept that has been elaborated on by various authors. Lund [13] showed how large-scale integration of wind energy could be performed in the case of different energy systems. Fitzgerald et al. [14] presented a case of integrating wind energy to the power system by efficient use of intelligent electric water heaters. Li et al. [15] investigated a stand-alone wind powered heat pump used for space heating. Beyer [16] investigated the meteorological conditions that might benefit the use of wind for heating purposes. Beyer and Niclasen presented a case study of matching the wind production with thermal load in the case of the Faroe Islands [17]. Mentioned studies have indicated theoretical potential, nevertheless these studies have left some unanswered questions. In general, wind generation coupled with a geothermal resource has already been considered [18], however not as supplementing sources, but instead by considering geothermal as the booster of a wind turbine generator [19].

Proposed solutions of wind power running resistive electric heaters for district heating purposes are also known and have been considered in the classifications [20].

In this work, the complementary operation of an undersized geothermal plant coupled with a wind farm and heat storage in an exemplary installation in Poland was investigated.

There were several serious rationales for this approach.

1.  Usually, wind turbines provide more energy during cooler periods (winter) (e.g., in Poland, see [7]). However, the lack of a good (preferably hourly) correlation with heat load implies a need for energy storage.
2.  Electric heaters work for all temperatures in district heating systems. In addition, they have reasonable investment and running costs.
3.  The utilisation of wind power to satisfy peak heat demand is a way to establish DH fully based on local resources, leading to almost 100% renewable, low maintenance heat supply to each and every dwelling within the range of the DH system.
4.  If wind turbines are not satisfying the demand, the deficits may be easily covered by buying energy from the national electric grid (if the system is operated in an on-grid mode).

In the present work, the authors would like to bring attention to the possibility of using wind power as a peak heat source for geothermal district heating. Wind can power electric heaters or heat pumps connected to voluminous insulated tanks used as a thermal storage.

The new contributions of this research paper are in the form of answers to the following research questions:

1)  Are undersized geothermal wells able to reliably provide heat for local district heating systems, if they are coupled with heat storage and a heat source powered by wind power?
2)  Is there only one unique solution for the given climatic conditions or a set of solutions?
3)  What is the minimum capacity of geothermal source heat power, wind power and heat storage capacity for given constraints? What is its performance over the period of multiple years (varying heat demand and wind speed conditions)?

## 2. Materials and Methods

To simplify the methodology, all given quantities were expressed as multiples of the heat demand under nominal conditions (100 W/m$^2$ to keep the interior 20 °C when the outside temperature was −20 °C, according to the Polish norms [21]). This rate would also be kept in other power designations, including wind: wind farm rated at "1" would be able to power the DH system with its nominal power equal to "1", etc. Units and their meaning are further elaborated in Appendix A.

The system should have been able to provide the required amount of energy for space heating for each and every hour. Heat demand comprised of space heating in houses and multistorey buildings and any other that was proportional to the external temperature. Sanitary hot water (SHW) preparation in the summer season was provided by the geothermal heat source. In winter, SHW preparation was included in the heat demand in space heating season. Please note, that low-temperature geothermal resources may represent temperatures sufficient for running medium temperature DH, so SHW preparation run solely by geothermal energy source was not excluded. Transmission losses of the distribution system were included in the heat demand.

The investigated system consisted of:

- geothermal heat source—power is mainly limited by water mass flow, temperature at the wellhead and the temperature of the DH supply power range is from 0 to 1 under the present consideration;
- heat storage—a container filled with hot water and electric heaters; its capacity varies in scenarios from 1 to 96 (1,6,12,24,48 and 96);
- wind farm—generates electric energy, its primary purpose is to keep the heat storage full, the surplus is curtailed in the off-grid operation mode; overall energy deficits can be covered by the national grid; installed power of wind varies from 0 to 1
- heat consumers—which generate heating demand in accordance with weather conditions. Their nominal demand is 1.

A scheme of the setup is presented in Appendix A.

The basic variables used were external temperature and wind speed at 10 m above the ground. External temperature was directly used to calculate the heat demand, while wind speed at 100 mabove ground was calculated from 10 m wind speeds by the use of a power law.

Capacities of both basic and peak heat source were limited (<DH peak demand). The objective was to minimize the rejected/curtailed wind energy, which for the dedicated operation would have to be wasted, causing (probably) economically unjustified operation. Thus, both for geothermal well and wind turbines the concept of avoiding oversizing is crucial. The excessive energy may be sold to the national grid in the on-grid operation mode, yet the operation of such hybrids is influenced by multiple factors (see Tables 1 and 2 in the Discussion and Results) and therefore for clarity this was not considered in this paper.

**Table 1.** SWOT (Strengths, Weaknesses, Opportunities and Threats) analysis on the potential of off-grid wind-geo district heating system development.

| STRENGHTS | WEAKNESSES |
|---|---|
| • the operation mode is relatively simple – wind generation is constantly used to keep thermal storage charged.<br>• there is no need to follow the grid code with regard to energy export from the wind farm.<br>• the capacity of lower wind generators can be easily replaced in case of malfunction/failure. | • by default there is a need to develop both a district heating and electricity transmission network.<br>• the potential of the system is determined by locally available wind resources.<br>• using wind potential from close by regions entails investment in own transmission network. |
| **OPPORTUNITIES** | **THREATS** |
| • price of heat produced for peak needs is easier to predict, independent from future fuels pricing | • the reliability of the system depends on a careful and precise design stage – the system cannot benefit from a potential backup from the national grid. |

**Table 2.** SWOT analysis on the potential of on-grid wind-geo district heating system development.

| STRENGHTS | WEAKNESSES |
|---|---|
| • the whole system can be undersized since in extreme conditions the additional energy supply can be provided by the national grid. | • a trade-off must be made between trading the electricity and supplying the heat demand – sometimes the market situation may indicate higher profitability of selling wind generation and compromising heat supply. |
| **OPPORTUNITIES** | **THREATS** |
| • the wind farm can be located far away from actual heat consumption and use the existing transmission network.<br>• the surplus wind generation theoretically will not be curtailed (depending on energy system situation) and will be exported to the grid.<br>• maximal available capacity factor can be reached as long as wind energy is accepted by the energy system.<br>• multiple operation modes are possible considering the use of wind energy – for example, wind can be used for thermal storage in such a manner that its ramp rates will be minimized. | • large wind farms cannot be usually located close to heat demand centres (urban areas).<br>• for large wind-geo projects network congestions may theoretically limit their operation capacity – especially in systems where heat demand is positively correlated with electricity demand. |

From the optimization perspective the problem had three decision variables, which determined the system parameters. Namely the capacity of geothermal source ($P^G$), capacity of wind source ($P^{WT}$) and capacity of thermal storage. The objective function (Equation (1)) was to minimize the wind curtailment whilst ensuring a certain level of system reliability (Equation (2)) and maintaining a certain level of the geothermal well utilization (Equation (3)).

Objective function:

$$\min\left(\sum P^{\text{WTr}}\right) \tag{1}$$

System reliability (LOLP – loss of load probability):

$$1 - LOLP \geq 0.95. \tag{2}$$

Maintaining geothermal capacity factor (CF) at a certain level:

$$P^{G\_CF} \geq Geo\_CF\_Min \tag{3}$$

The algorithm of the performed operations is presented in Figure 1 and later elaborated in detail in Appendix B.

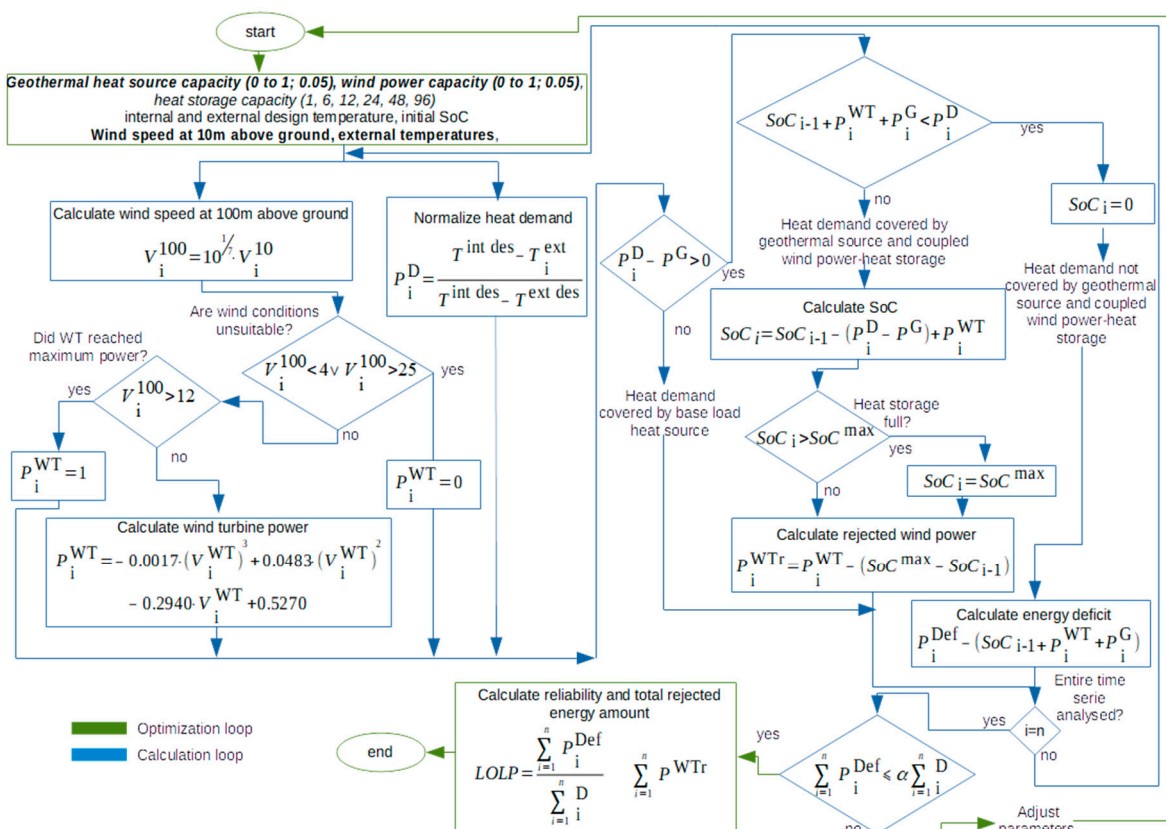

**Figure 1.** The procedure of calculation performed during the modelling process. The parameters that were changed in consecutive optimisation loops are in ***bold italic font***, constant parameters are in *italic font*, presumptions and descriptions are given in normal font, whilst **bolded** are input data in time series. The parameters are given in brackets (value range; increment). The algorithm of the optimisation loop is elaborated in Appendix B. Symbols, superscripts and subscripts are explained in Abbreviations.

## 3. Results and Discussion

For the purpose of method validation, the simulation was performed based on a typical meteorological year (TMY) for Warsaw (and later tested for resilience based on a 35 year-long hourly time series obtained from a dedicated database [22]). Primarily, 95% RES (renewable energy sources) reliability (understood as a ratio between total energy provided by RESs and total energy demand) was assumed along with a 40% capacity factor of the base heat source (geothermal well), which was typical for Poland [8]. The authors would like to emphasize that results were representative for the chosen location and presumed parameters, albeit the model may have easily been adapted for other conditions.

Figure 2 presents all acceptable solutions within the given constraints and for the heat storage equal to 96 units. A minimum amount of wind power was rejected when the geothermal heat source was small or none. However, most of such potential system configurations did not provide required RES reliability (minimum 95% for TMY) and were excluded. RES reliability of acceptable solutions is represented by the colour of the dot on top of each bar.

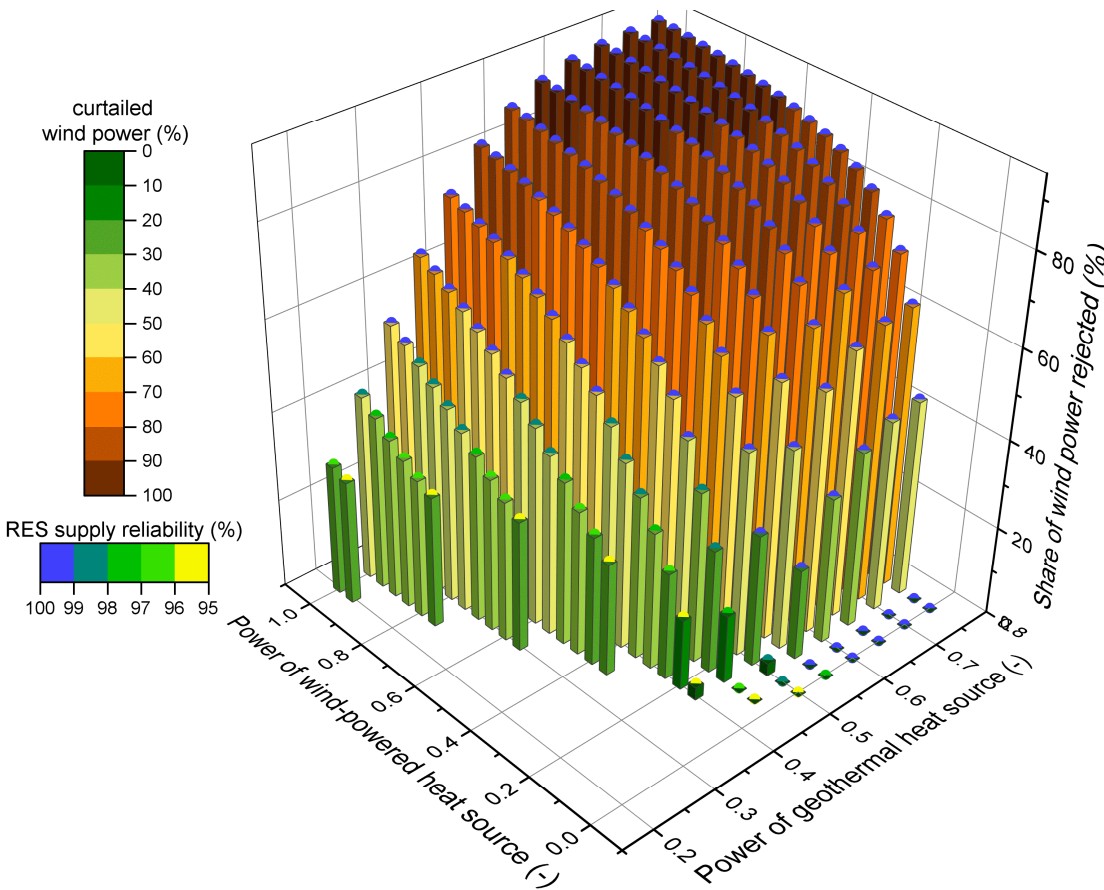

**Figure 2.** Curtailed wind power and reliability of energy supply from renewable energy sources for various system configurations for a typical meteorological year. The heat storage capacity is equal to the energy needed to satisfy the energy demand for 96 hours of nominal district heating (DH) operation.

The base source's capacity factor reached 57.6%, which was a satisfying result (assumption: temperature provided by the geothermal source is high enough to be fully useful during the entire year). Mean CF calculated by production for six geoDHs in Poland in 2015 was about 34% [8]. This indicated two factors: temperature levels accordance and energy demand distribution over the year were the main limiters of the geothermal heat source capacity factor. Thus, low- and ultra-low temperature district heating systems facilitate the use of such hybrid systems.

For the smallest setup configuration, which satisfied the objective function and it's constraints for the storage of 96 units, over 95.8% of heat demand was satisfied by renewables over the typical meteorological year. As was expected, adding even a small amount (0.05) of wind power allowed for the limitation of the geothermal well capacity to 0.45. Smaller heat storage (1.0) required more wind power (0.15), whilst the smallest wind power was needed when heat storage excessed 48 units of energy (reminder—one unit is distributed during one hour of nominal DH power). When wind power did not exceed 0.05, no curtailment took place, regardless of the geothermal plant power capacity. Additional simulations (not visualised here) indicated that the geothermal source's capacity factored higher than 60%, which was also possible, both for smaller and bigger heat storage capacities, nevertheless, it resulted in some wind power curtailment. What was not surprising, was that

capacity factors as high as 80% were not available, which was due to the lack of heat demand in the summer season.

One of the most vital aspects of the economic and environmental performance of the proposed system is the amount of wind power curtailment. Assuming an off-grid wind farm operating solely for DH purposes and used as a peak heat source, produced wind energy may be utilised only if there is a need to charge thermal storage or the geothermal source is not capabe of covering the heating load alone. Thus, it is crucial to investigate what share of wind power would be curtailed. Analyses have been performed for more than 35 years using historical temperature and wind speed data. Results regarding excessive wind power production are shown in Figures 3 and 4. It can be observed in Figure 3 that neither the size of the wind power nor heat storage strongly influenced the share of wind power curtailed during the investigated period of 35 years. Instead, the rate of curtailment mostly depended on the year considered. Power from the wind farm was not always produced in accordance with the heat demand—there were significant differences within the presented sequence of years. There were years in which an increase of storage capacity only slightly modified the share of surplus wind power in its total production (eg., 1989, 1995), whilst there were years for which this modification was evident (eg., 2005, 2011; best visible in the red set of lines).

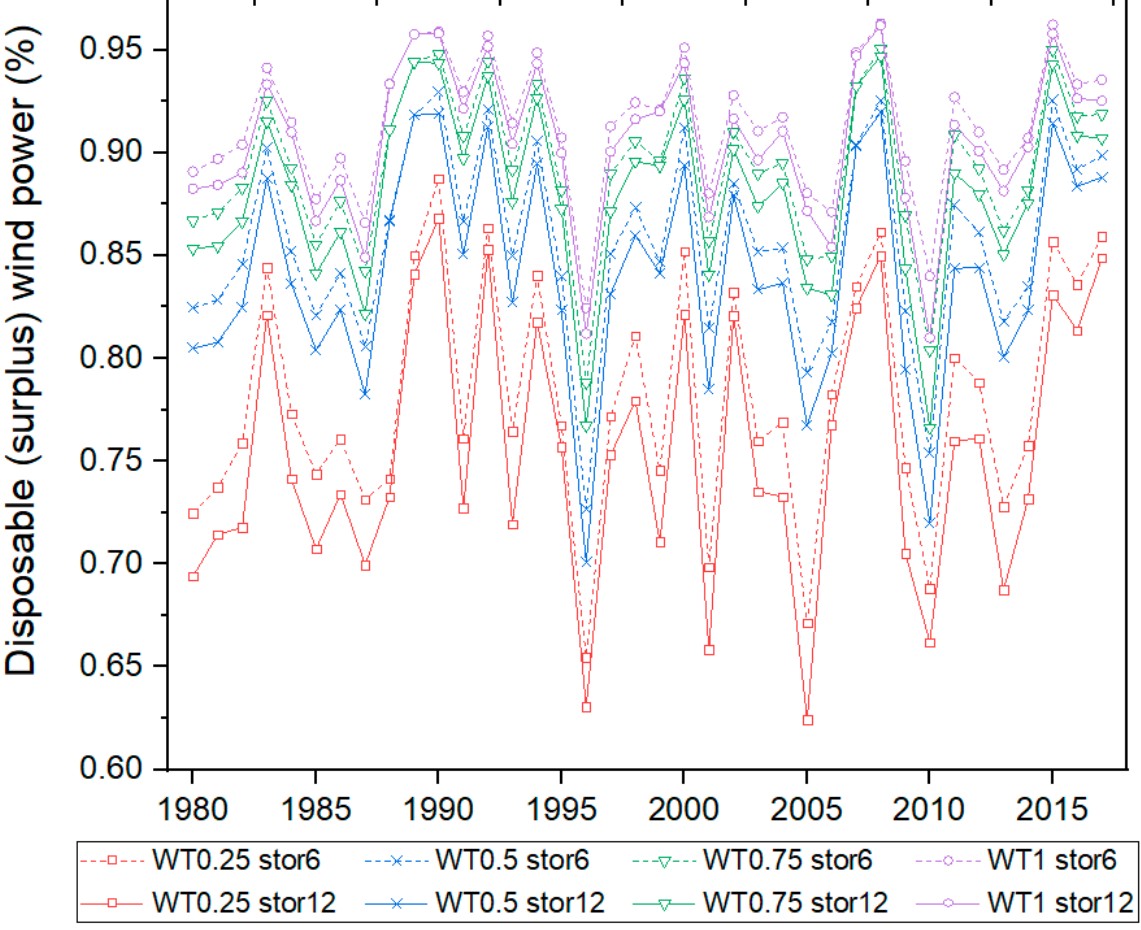

**Figure 3.** The share of surplus wind power (energy rejected) for various configurations of wind power and heat storage over 37 consecutive years of observation. The capacity of the geothermal source equalled 0.5. Abbreviations used in the graph: WT–wind turbines capacity; stor–heat storage capacity. Explanation of labels: WT0.25 stor6 means that the considered system consisted of a geothermal heat source of size 0.5, heat storage with capacity of 6 units and a wind farm capacity of 0.25. Units used in the graph are extensively explained in Appendix A.

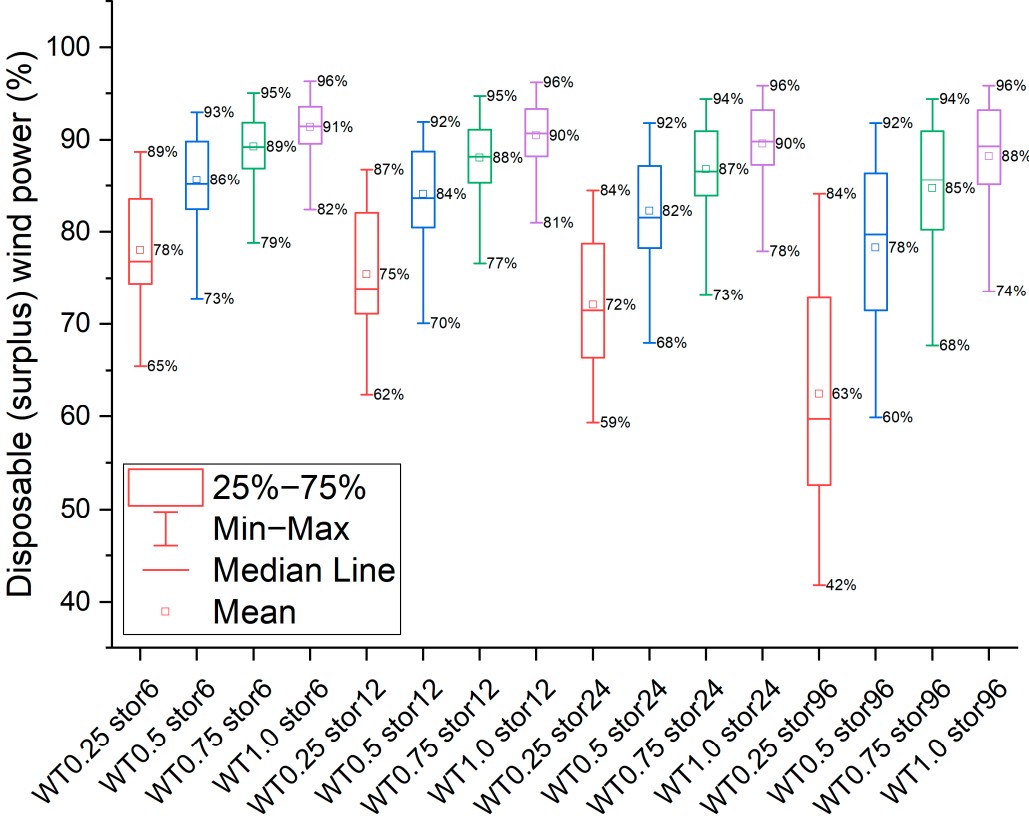

**Figure 4.** Ranges of curtailed wind power for various configurations of wind turbines and heat storage sizes over 37 consecutive years of observation. Geothermal source capacity equals 0.5. Labels explanation–see Figure 3. Units used in the graph are extensively explained in Appendix A.

In general, an increased storage capacity decreased the share of excessive wind power, although the proportional reduction was not the same for every year, whereas an increased installed wind power always resulted in an increased share of curtailed wind power and this modification was generally proportional over the years.

An interesting observation that underlines the variable character of onshore wind resources was that increasing storage capacity did not significantly limit the amount of wind power curtailment, instead, the ranges became wider with the upper limit being almost constant and a modified lower limit. This may be explained by the occurrence of years in which wind and temperature conditions were incoherent in short time periods (eg., comparatively short periods of wind resource abundance, when the storage got full with minor discharge followed by a time of moderate or low wind potential when energy was consumed). Unsurprisingly, the increase seen in the installed wind power capacity led to an increase in curtailment. Even increasing storage capacity by a factor of 16 (from 6 to 96, see Figure 4) reduced the share of curtailment by only a 3 to 15 percentage point. The amount of curtailment can be used as an indicator to calculate wind power cost for the dedicated operation of the wind farm or the amount of disposable energy, which may be sold to the national grid.

Two approaches seem economically justified for the combination of geothermal heat, wind energy and district heating. The first option is an off-grid wind farm, which would possibly be small and use most of its production for heating and storage purposes. The remainder would be curtailed or used to cover the electrical energy needs (if a multienergy system was considered).

Curtailment itself is not in line with sustainability, as decreased utilisation of produced energy increases the $CO_2$ footprint of a used energy source – the production and construction of a wind farm entails the same emissions, regardless of the amount it produces. Storage might be optional,

as it may be more expensive than curtailing energy. This is a site-specific question, which is worth further investigation.

The second option is to establish an on-grid wind farm operating for most of the time and selling the major share of its production to the grid. Such a solution may be competitive compared with for example natural gas peak sources, which also require maintenance and significant investment costs while being used only for a very limited time of the year. The first solution requires heat storage of a larger capacity and moderate dimensions, while the second one requires a smaller storage capacity and a larger wind farm dimension.

For the considered location of Warsaw (Poland) and heat storage of 96 units, a solution for the first approach would be to have a wind farm with a rated power of 0.05 in the given units and a heat power of the geothermal facility of 0.45. Such a system configuration satisfies the constraint of 95% RES supply reliability with both wind and geothermal power, while at the same time being of minimal capacity. For the optimal size of the second possible approach, additional calculations would be required, as economical aspects affect the optimisation process. The SWOT (strengths, weaknesses, opportunities and threats) analyses for both options (on and off-grid) was performed and is presented in Tables 1 and 2.

The results presented in Figure 5 indicate the real opportunity of running a geo-WT DH system, where wind increases the renewable share significantly and in rare cases of RES insufficiency the energy deficit would be covered by the national grid.

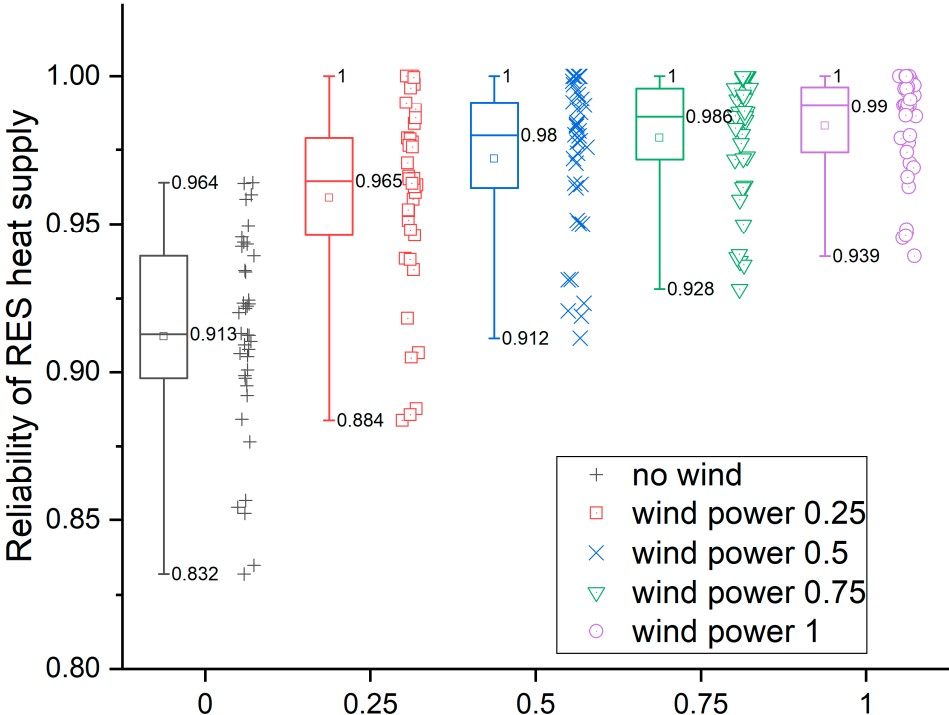

**Figure 5.** Reliability of energy supply from renewable energy sources depending on the size of the wind power farm. The system consisting of heat storage of 6 units capacity and a geothermal heat source of 0.5 unit capacity. Units used in the graph are extensively explained in Appendix A.

Although wind farm performance is important, the parameters of geothermal source exploitation is equally vital. For the proposed system and configuration described in the caption of Figure 6, high capacity factors may be obtained. It is economically beneficial and has the potential of improving maintenance and reservoir exploitation conditions. High shares of heat demand covered solely by the geothermal heat source show a significant potential for improvements in the economic performance of the geothermal well if its capacity is undersized. Simultaneously, the constant rate of geothermal reservoir exploitation increases the safety of the reservoir, which may be damaged by rapid pumping

rate changes. As the algorithm was set to use the geothermal source as source one, the obtained CF values were the same for different wind power capacities.

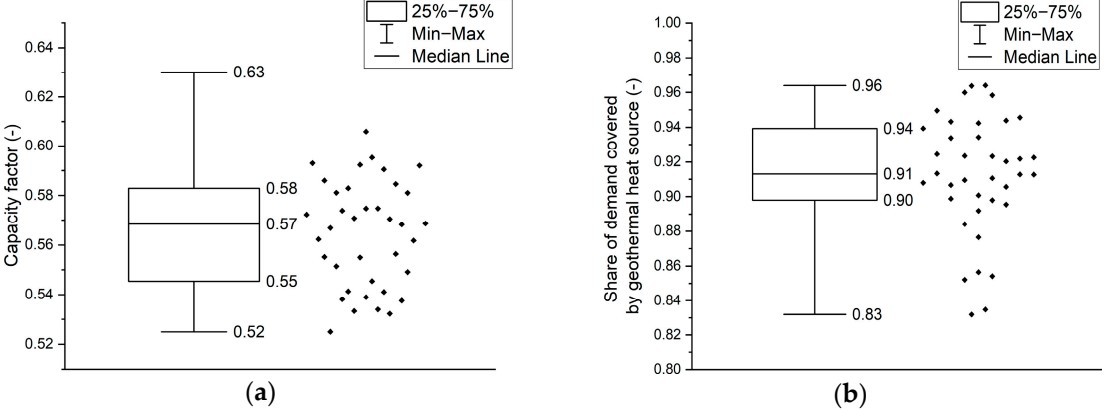

**Figure 6.** (**a**) Capacity factor for geothermal source and (**b**) share of energy demand satisfied by geothermal energy. The system consists of a geothermal heat source of 0.5 unit capacity, heat storage of 6 units capacity and wind power of any capacity.

For comparison, a simulation for 99% reliability was also performed. Under the same conditions, geothermal source capacity factors decreased, although they were in a broader range. The capacity of the geothermal source was required to be higher at 0.6. However, it decreased significantly with sufficient wind power (0.3–0.8) as well as heat storage (48–6 DH operation hours at nominal load, respectively). This meant that a wind source with a capacity of 0.3 would require eight times larger storage than the one with 0.8 capacity.

One of the main constraints in the present article is the reliability of the proposed solution. The reliability of heat production may be 100%, in spite of comparatively low percentages presented in Figure 5, as missing energy may be bought from the national grid. Then, a decision on operation mode needs to be made (WT separated from the grid or typical on-grid operation). For concrete decision-making an optimisation including detailed economic conditions is necessary. The Strengths, Weaknesses, Opportunities and Threats (SWOT) analysis (Tables 1 and 2) presented below gives an insight into both potential solutions.

## 4. Conclusions

From our findings, we drew the following conclusions:

- Wind energy is a suitable complement for geothermal heat at times. However, further economic analysis is necessary to deepen the understanding of the use cases.
- In all cases, significant amounts of curtailment seem inevitable. Even increasing the size of the heat storage does only partially reduce this problem.
- Different weather years show strongly variable results. This indicates that proper planning of a wind-supported geothermal district heating system should include the local climatology of the considered site.

Regarding the research questions that were presented in the introduction, we found out that:

1) An undersized geothermal well coupled with a wind farm is able to reliably provide heat for a district heating system.
2) There are multiple combinations of geothermal well, wind turbine and heat storage proportions that provide the required reliability over a typical meteorological year; without additional constraints (like economical conditions or reliability over a long time period) it is impossible to indicate the optimal solution.

3)　There is a minimum ratio for the given constraints in the climate conditions of Warsaw. For the heat storage capacity of 96 units the geothermal heat source should be 0.45 and wind source 0.05. If the theoretical geoDH-wind network was supplying 1000 m$^2$ of households, which require 100 W/m$^2$ when the outdoor air temperature is −20 °C then following our assumptions (Appendix B) the system should have the following capacities: 4.8 MWh in thermal storage, 45 kW in geothermal source and 5 kW in wind source.

## 5. Future Research Directions

For future work, we have distinguished the following potential research directions:

- To include thermal insufficiency of the geothermal source as a reason for peak source use.
- To develop the necessary methodology and perform a similar assessment for entire regions or countries (Poland).
- To estimate the conditions under which energy storage is less beneficial (financially and ecologically) than wind power curtailment.
- To explore the potential in spatial complementarity of wind sources in areas surrounding already existing geothermal district heating systems in Poland.
- To modify the algorithm in order to decide whether to use wind power for charging thermal storage or selling it on the energy market (for an on-grid system).
- To modify the algorithm to charge the heat storage with surplus power or only during the night (considering the current state of the energy system).

**Author Contributions:** Conceptualization, B.C., J.J.; model design B.C, J.J.; model calculations J.J.; writing—original draft preparation, B.C.; data visualisation, B.C.; discussion B.C., J.J. and A.K.; writing—review and editing, A.K., J.J.

**Funding:** This research received no external funding.

**Acknowledgments:** Bartłomiej Ciapała would like to express his gratitude to Kazimierz Broniszewski for the inspiration. Jakub Jurasz acknowledges the financial support of the Foundation for Polish Science (Pol. Fundacja na rzecz Nauki Polskiej, FNP). Alexander Kies was financially supported by Stiftung Polytechnische Gesellschaft and the R&D project NetAllok (FKZ03ET4046A), financed by the Federal Ministry of Economic Affairs and Energy (BMWi).

**Conflicts of Interest:** The authors declare no conflicts of interest.

## Abbreviations

| | |
|---|---|
| CF | capacity factor |
| DH | district heating |
| Geo_CF_Min | minimal allowed geothermal source capacity factor |
| GeoDH | geothermal district heating |
| LOLP | loss of load probability |
| LTDH | low-temperature district heating |
| ULTDH | ultra-low-temperature district heating |
| P | power |
| RES | renewable energy source |
| SHW | sanitary hot water |
| SOC | state of charge |
| T | temperature [K] |
| TMY | typical meteorological year |
| V | wind speed [m/s] |

**Sub- and super-scripts**

| | |
|---|---|
| D | demand |
| Def | deficit |
| G | geothermal |
| i | any hour during considered period |
| Max | maximal |
| n | last hour of the considered period (number of instances) |
| S | supply (provided) |
| G_CF | geothermal capacity factor |
| WT | wind turbine |
| WTr | wind power rejected |
| $\alpha$ | coefficient (one minus reliability) |
| int des | internal designed |
| ext | external |
| ext des | external designed |

## Appendix A. On Unitless Approach for System Sizing

Figure A1 visualizes the conceptual desing of a geo-wind system with thermal storage. The system can operate in both on and off-grid modes.

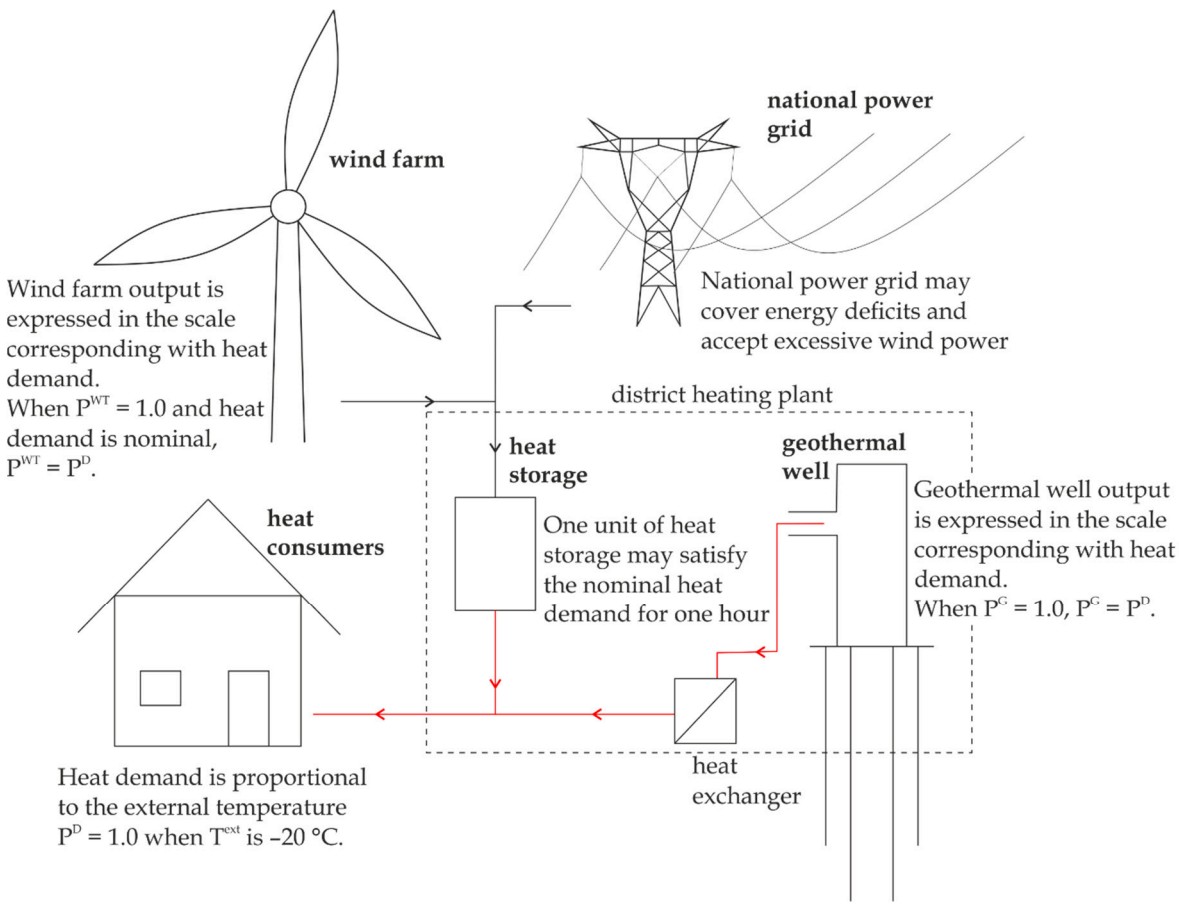

**Figure A1.** Conceptual design of the geo-wind system with corresponding heat and electricity flow.

The basic dimension is heat demand generated by heat consumers. It is equal to 1, when the external temperature is −20 °C. It may be understood as 1 $MW_t$ or an entire DH nominal demand, eg., 3.6 GJ/h. Example: during hour *i* the heat demand was 1.0, which meant that heat consumers required 3.6 GJ, but in hour *i*+1 demand was 0.5 so the heat demand was 1.8 GJ.

Heat demand was covered by a geothermal well or heat storage. The thermal output of the geothermal well is expressed in the same units as DH, yet may be smaller than DH itself (a case of an undersized geothermal source).

Heat storage is powered by wind turbines or a national grid (depending on the assumed operation mode, on/off-grid). Power is transformed to heat by electric heaters, which deliver heat at any temperature level required by the DH. Nominal power of the wind turbine was 1.0, when its nominal production transformed losslessly to heat, which may have satisfied a heat demand equal to 1.0. In reality, WT power may be expressed in MWe, kWe or other units relevant for electricity generators.

**Table A1.** An example of how the approach to units was presented in this study including how it should be converted to real case dimensions. For example the maximal heat demand observed in DH was 3 MW. It was represent by "1" in the sizing method. The optimal system configuration was 0.5 in the geothermal source, 0.1 in te wind turbine and the storage capacity was 96 units. When multiplied by "3MW", the following (third row) parameters were obtained.

| DH | Geothermal Source | Wind Turbines | Heat Storage |
|---|---|---|---|
| 1.0 | 0.5 | 0.1 | 96 |
| 3 MW | 1.5 MW | 300 kW | 288 MWh |

## Appendix B. On Optimization Procedure That Was Applied

In linguistic form the optimization problem of geo-wind district heating system, reads as follows: Given the heat demand, which has to be covered with a certain loss of load probability, this can be covered by the geothermal source (baseload) and wind turbines. Electricity from the wind is converted to heat via resistive heaters. In addition, the possibility of storing heat exists. Under the given constraints that loss of load probability must be below a certain threshold and the capacity factor of the geothermal energy source must be above a certain threshold, first the minimal geothermal capacity is determined and in a second step the minimal wind capacity. This process is repeated for different scenarios of heat storage capacity (1, 6, 12, 24, 48 and 96 hours of the typical demand of the system). This procedure was implemented in MS Excel and operation on matrices were performed (following an exhaustive search/brute force approach) and is visualised in the figure below:

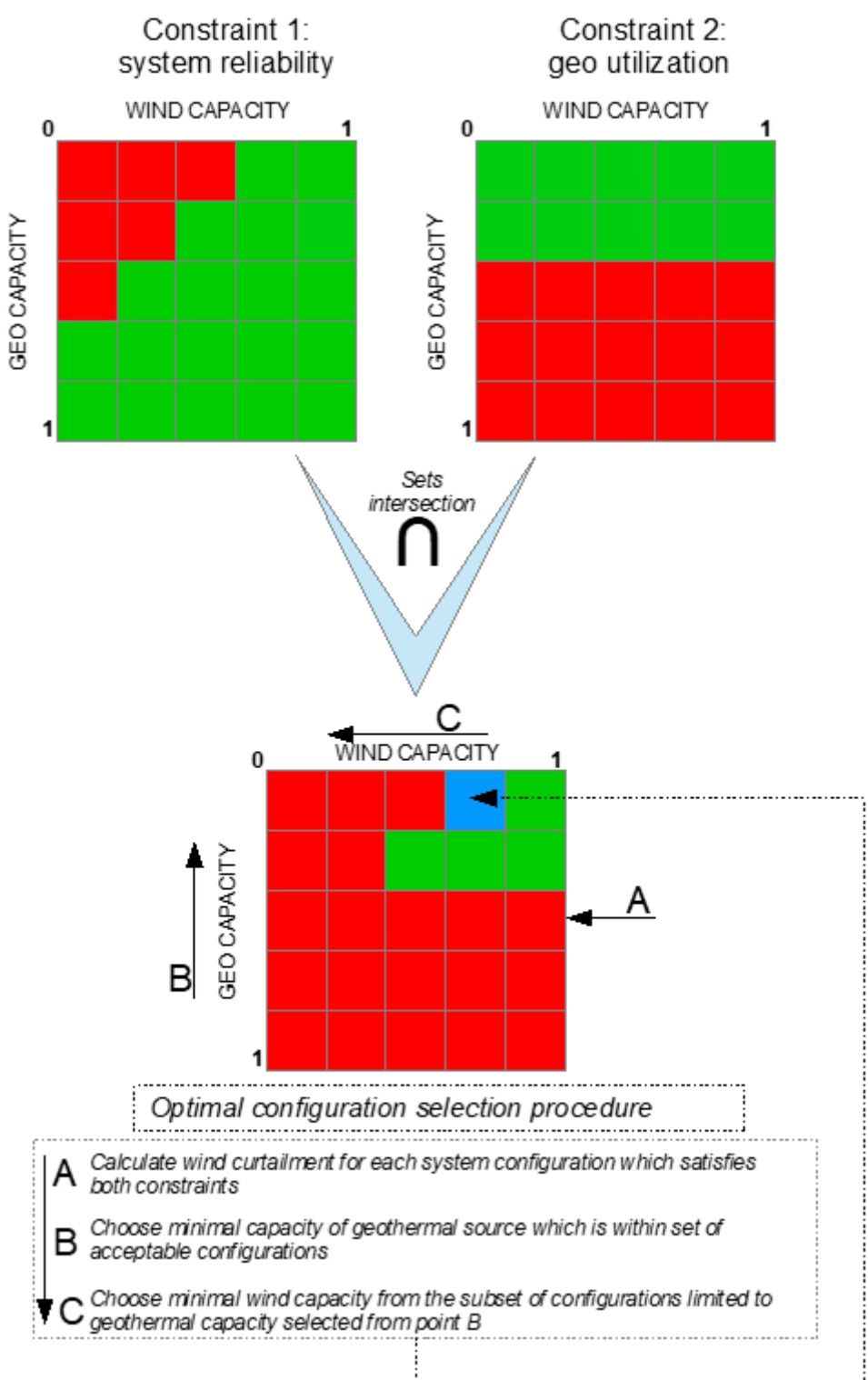

**Figure A2.** Optimization procedure implemented in this research.

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
