# Peer review of "The Potential of Wind Power-Supported Geothermal District Heating Systems—Model Results for a Location in Warsaw (Poland)"

_energies, doi:10.3390/en12193706_

Round 1
Reviewer 1 Report
This paper by Ciapala et al. study, by modelling, a ‘local, conceptual’ district heating system consisting of a geothermal heat source, local heat storage and wind power generators using historical weather data of Warsaw (Poland). As is usually the case for geothermal plants, due mainly to economic constraints (very expensive wells), the geothermal part is ‘undersized’ and need be supplemented to cover ‘peak demands’ or even longer periods, e.g. during wintertime.
This concept of combining district heating from a geothermal source with wind power and heat storage is clearly of great interest. This may ensure a district heating system fully based on local resources and leading to almost 100% renewable energy supply, as emphasized by the authors.
The idea and the aim of study (see below) is thus clearly of great general interest. Unfortunately, is both the description of methodology and the results, as presented, not at a stage where it is ready for publication.
The methodology, in terms of an optimization procedure, is very briefly described with the theoretical relationships, outlined in Fig. 1, however, basically, with no explanation. Methodology, including theoretical relations and procedures need explanation, not just indicated in a figure. Parts should conveniently be outlined and explained in an appendix.
Modelling results are presented mainly in figures, however, generally without explanation and with short comments here and there in the main text. Importantly, the results and general conclusions are not sufficiently aligned with the main aim of the study.
The aim of study is quite clear, as outlined by the authors (starting line (L) 79):
“The novel aspects of the present elaboration aim at answering the following researching questions:
1) Are undersized geothermal wells able to reliably provide heat for local district heating systems, if they are coupled with heat storage and a heat source powered by wind power?
2) Is there only one unique solution for given climatic conditions or a set of solutions?
3) What is the minimum ratio of geothermal source heat power, wind power and heat storage capacity for given constraints? What is its performance in conditions over multiple years?”
From the conclusions (starting L 248) and the text in general, these questions are not clearly answered:
“From our findings, we draw the following conclusions:
Wind energy is a suitable complement for geothermal heat at times. However, further economic analysis is necessary to deepen the understanding of the use cases.
In all cases, significant amounts of curtailment seem inevitable. Even increasing the size of the heat storage does only partially reduce this problem.
Different weather years show strongly varying results. This indicates that proper planning of a wind-supported geothermal district heating system should include the local climatology of the problem site.”
Locally in the text, we find element of results such as: L 149:
“For the smallest setup configuration which satisfies the objective function and it’s constraints for 96-units storage, over 95.8% of heat demand is satisfied by renewables over the typical meteorological year. Expectably, adding even a small amount (0.05) of wind power allows limiting the geothermal well capacity to 0.45. Smaller heat storage (1.0) requires more wind power (0.15), whilst the smallest wind power is needed when heat storage excesses 48 units of energy (reminder - one unit is distributed during one hour of nominal DH power). When wind power does not exceed 0.05, no curtailments of takes places, regardless of the geothermal plant power capacity.”
However, details of normalized numbers are very difficult to follow due to lack of explanation and main findings are not clearly outlined. To me, a number of abbreviations are not sufficiently introduced, which makes it even more difficult to follow.
The above contains my main criticism.
Furthermore, a number of specific items. Just some few examples.
The system, which is emphasized, seems to be ‘an off-grid wind farm’, rather than (L 208) “an on-grid wind farm operating for most of the time and selling the major share of its production.”
Here, some inconsistencies seem to exist:
L 106: “wind farm - generates electric energy, its primary purpose is to keep the heat storage full, the surplus may be sold to the national grid; overall energy deficits are covered by the national grid; installed power of wind varies from 0 to 1”
And:
L 119 “The objective is to minimize rejected wind energy, which for the dedicated operation would have to be wasted, causing economically unjustified operation. Thus, both for geothermal well and wind turbines the concept of avoiding oversizing is crucial.”
Thus, is the ‘surplus energy’ wasted or ‘sold to the national grid’? This is not clear and important for the optimization (and economy).
Actually, to me, the more interesting option, of ‘an on-grid wind farm’, seems hardly analyzed/discussed. It is commented as (L 217): “For the optimal size of the second possible approach (me: the on-grid wind farm), additional calculations would be required, as economical aspects affect the optimisation process.”
Clearly, also for the current system, with the ‘off-grid wind farm’, ‘economical aspects will affect the optimization process’.
To me, before ready for publication, much work is needed, both generally in terms of structuring and presenting methodology and results to be understood by the reader and related to specific items, some of which are mentioned above.
Reviewer 2 Report
Description of the investigated system should include units. It is unclear what capacities are investigated.
There is no information about the type of geothermal source (vertical/horizontal, open/close...)
Graphs should contain units
In general opinion is that high potential energy (electricity) is very valuable product , and use of it for production of energy with lower potential (heat) should be considered only in situation when electrical grid can not utilise all the electrical energy at certain point of time (during off peak hours at night, specific weather (wind, water oversupply) condition etc.). Primary use of wind farm for keeping heat storage full is not the most attractive termodinamical decision. It is more attractive to utilise electricity as much as possible for the electrical appliances and only excess electricity could be used for heating.
Round 2
Reviewer 1 Report
The manuscript has been significantly improved by considering my comments and suggestions.
Parameters, and algorithms as well as abbreviations are better explained in particular by the improvements and additions in appendixes. Also the improvements of the SWAT analysis is important.
Clearly, there is now a better consistency between research questions, abstract and conclusions.
Some improvements of the English language and style seems needed.
Information on place of publication/journal for refs. 1, 2 and 18 are not given.
Author Response
Thank you for your comments. We have revised the manuscript accordingly to your suggestions.
Reviewer 2 Report
/
Author Response
Thank you for accepting our responses and the revised version of the manuscript.